# Mental Health and Quality of Life of Women One Year after Maternal Near-Miss in Low and Middle-Income Countries: The Case of Zanzibar, Tanzania

**DOI:** 10.3390/ijerph17239034

**Published:** 2020-12-03

**Authors:** Sisay Alemu, Tanneke Herklots, Josue Almansa, Shadya Mbarouk, Esther Sulkers, Jelle Stekelenburg, Janine de Zeeuw, Benoit Jacod, Regien Biesma

**Affiliations:** 1Global Health Unit, Department of Health Sciences, University Medical Center Groningen, University of Groningen, 9713 GZ Groningen, The Netherlands; e.sulkers@umcg.nl (E.S.); j.stekelenburg@umcg.nl (J.S.); r.biesma@umcg.nl (R.B.); 2Department of Obstetrics and Gynaecology, Erasmus MC, University Medical Centre Rotterdam, 3015 GD Rotterdam, The Netherlands; tannekeherklots@gmail.com; 3Department of Health Sciences, Community and Occupational Medicine, University Medical Center Groningen, University of Groningen, 9713 GZ Groningen, The Netherlands; j.almansa.ortiz@umcg.nl; 4Department of Obstetrics and Gynaecology, Mnazi Mmoja Hospital, Zanzibar, Tanzania; khasam.mb@gmail.com; 5Department Obstetrics & Gynaecology, Leeuwarden Medical Centre, 8934 AD Leeuwarden, The Netherlands; 6Department of Health Sciences, University Medical Center Groningen, University of Groningen, 9713 GZ Groningen, The Netherlands; j.de.zeeuw@umcg.nl; 7Department of Medical Sciences, University Medical Center Groningen, University of Groningen, 9713 GZ Groningen, The Netherlands; 8Department of Obstetrics and Gynaecology, Onze Lieve Vrouwe Gasthuis, 1091 AC Amsterdam, The Netherlands; bjacod@gmail.com

**Keywords:** maternal near-miss, mental health, quality of life, trajectories, Sub-Saharan Africa, Zanzibar

## Abstract

Women who experienced a maternal near-miss are at risk of mental health complications and lower quality of life, but long-term consequences are largely unknown. The aim of this study is to assess whether mental health symptoms and quality of life change over time and to examine associations with risk factors among post-partum women. In this cohort study, women with maternal near-miss were matched to women without or with mild complications at Mnazi Mmoja Hospital in Zanzibar. Depressive and post-traumatic stress disorder symptoms, and quality of life were measured at three, six, and twelve-months follow-up. A linear mixed-effects model was used for data analysis. Postpartum women in Zanzibar reported low levels of depressive and post-traumatic stress disorder symptoms. While depressive symptoms and quality of life trajectories were similar among women with and without maternal near-miss, differences for trajectories of post-traumatic stress disorder symptoms and physical quality of life were found. Social support, perinatal loss, and intercurrent illness were strongly associated with both depressive symptoms and quality of life in this group of Islamic women. These findings suggest that social support, embedded in the cultural context, should be considered in helping women cope with mental health issues in the aftermath of severe maternal complications.

## 1. Introduction

While maternal mortality ratios are declining globally, women increasingly experience life-threatening obstetric complications particularly in low and middle-income countries (LMICs) [1]. For every maternal death in LMICs, nearly 30 women survive a life-threatening obstetric complication; hereafter referred to as maternal near miss (MNM) [2]. Care for women after MNM usually ends at discharge from the hospital [3]; nevertheless, the consequences of MNM can be long lasting. The impact is usually multidimensional, affecting physical functioning, mental health, and socio-economical aspects [3,4]. So far, some studies researched long-term mental health consequences in women with MNM, such as depression and quality of life (QoL), but results are inconclusive [5,6,7,8,9,10,11,12,13]. For instance, most studies (e.g., conducted in Morocco [7], Benin [8], and Burkina Faso [9]) found that women with MNM had an increased risk of developing depression in the post-partum year, especially if they had lost their baby. On the contrary, similar studies in Malaysia [10] and England [5] found no significant difference regarding the risk of depression among women with and without MNM. Similarly, there is contradicting evidence regarding the association between hospital admission due to pregnancy complications and post-traumatic stress disorder (PTSD) [5,6,11]. This study aims to investigate mental health (depression and PTSD) and QoL of these women over time.

## 2. Materials and Methods

### 2.1. Study Design, Data Sources, and Population

In this prospective case-control cohort, all women who were pregnant or within 42 days of termination of pregnancy and admitted to Mnazi Mmoja Hospital between 1 April 2017, and 31 December 2018, were eligible. Mnazi Mmoja Hospital is the referral hospital of Zanzibar, a semi-autonomous archipelago of Tanzania. It accommodates about 30% of all facility deliveries in Zanzibar, with around 13,000 deliveries per year. The in-hospital maternal mortality ratio is around 400 per 100,000 livebirths. MNM cases were identified using locally validated version of the World Health Organization (WHO) MNM criteria [14], with adjustment on some criteria to fit the local setting (see Appendix A). Each MNM case was matched with a control, a woman with no or only mild complications, on the following variables: Age, mode of delivery, termination of pregnancy, and time of admission. Excluded were those patients: (1) without a mobile phone, (2) residing outside of Unguja Island, and (3) with acute psychiatric disorder compromising obtaining informed consent or pre-existing psychiatric disorder potentially confounding any outcomes. A research assistant invited all eligible women in the hospital during the study period who fulfilled inclusion criteria to participate in the study. Inclusion in the study took place after written (or oral in case of illiteracy) informed consent was obtained from each participant. Baseline characteristics were collected before discharge by the same research assistants through structured interviewed. Follow-up was performed at three, six, and twelve months after discharge, either at the hospital or at home according to participant’s preference. All research assistants were female Zanzibari with a bachelor’s degree in counseling psychology and fluency in English and Kiswahili. All assessments were conducted in Kiswahili.

### 2.2. Outcome Variables

The primary outcome variables were depressive symptoms measured using Patient Health Questionnaire-9 (PHQ-9), post-traumatic stress disorder (PTSD) symptoms using the Harvard Trauma Questionnaire-16 (HTQ-16) and QoL using WHO’s quality of life scale (WHOQOL-BREF).

PHQ-9 is a depression symptom scale, which rates each of the nine Diagnostic and Statistical Manual (DSM-IV) criteria on a four-point Likert scale. PHQ-9 scores of 5, 10, 15, and 20 represent mild, moderate, moderately severe, and severe depression, respectively [15]. The PHQ-9 has been translated and validated in Tanzania [16]. The HTQ-16 questionnaire screens for PTSD using 16 trauma symptom items derived from the DSM-IV PTSD criteria, rating each item on a four-point Likert scale. The HTQ total score is an average score, with 2.5 and above indicating a high likelihood of PTSD [17]. The HTQ-16 has been validated in SSA [18] and used in cross-cultural research [17,18,19]. The WHOQOL-BREF, a shortened version of the WHOQOL-100, is applicable in situations with time constrains and respondents’ burden must be minimized. WHOQOL-BREF has been validated cross-culturally [20]. In this study, each domain of QoL (physical health, psychological health, social relationships, and environmental domain) was used as a separate outcome variable.

All questionnaires, except PHQ-9, were translated by three research assistants, who are health-care workers and fluent both in Kiswahili and English. Translation was performed using the World Bank three step questionnaire translation: (1) Forward translation (one research assistant translated the original scale to Kiswahili), (2) back translation (a different research assistant translates the Kiswahili version back to English), and (3) reconciliation (the three research assistants together compared the original scale to the back translated questionnaire, discuss all discrepancies and agree on a final translation) [21].

### 2.3. Risk Factors

Baseline characteristics used as factor were: (a) Pregnancy outcome categorized as follows: Livebirth (child alive at discharge, at least one in case of multiples, or the mother is still pregnant during discharge), perinatal loss (intra-uterine fetal death or neonatal death of all babies), and early pregnancy loss; (b) mode of delivery, i.e., vaginal delivery, caesarean section or early pregnancy loss; (c) gestational age at the end of pregnancy or occurrence of MNM if still pregnant at discharge; (d) parity; and (e) history of caesarean section. In addition, perceived social support was measured using the social support domain of the Hopkins Symptom Checklist-25 (HSCL-25). Also, participants were asked at each follow-up moment about intercurrent illness defined as a medical consultation in an outpatient clinic or health center in the period prior to the interview.

Possible confounders included were: Participant’s age, socio-economic status (measured on the basis of employment, educational level, and perceived wealth (below average, or average or higher)), residence (urban, rural, or mixed), and marital status. Most independent variables were measured only at baseline, however social support and intercurrent illness of the mother were measured at three, six, and twelve-months follow-up because they were considered as possible time-varying predictors.

### 2.4. Missing Data

When a maximum of two items per questionnaire (per domain in case of WHOQOL-BREF) were missing, they were imputed with the mean score of remaining items in that questionnaire (or domain in case of WHOQOL-BREF), except for the smaller social relationships domain in WHOQOL-BREF where the maximum acceptable missing items was one [22]. Missing values were assumed to be missing at random (MAR) [23].

### 2.5. Statistical Analysis

Mixed models were used to estimate the trends of the outcomes over time (centered at 3 months), and to explore if they differ between near-miss and control groups we included interactions between time and group. QoL domain outcomes were right-skewed with a strong ceiling effect; thus, we modelled the reversed of QoL scores as zero-inflated negative-binomial distribution. The zero-inflated negative-binomial distribution includes two types of parameters: The negative-binomial parameters, which refer to the expected average score, and the zero inflated parameters, which refer to the probability of having an excess of the ceiling score (100). Depressive symptoms were left-skewed, thus analyzed as a negative binomial. Given that PTSD symptoms had a very irregular distribution with few HTQ-16 values greater than 1, we categorized them as binary (one and more than one) and analyzed as longitudinal logistic model.

Covariates for QoL domains and depressive symptoms were added using forward stepwise method. We could not identify associated factors of PTSD symptoms because of the irregular distribution. Mixed models were estimated in R software version 3.6.1 [24].

### 2.6. Medical Ethical Considerations

The study was approved by the Zanzibar Medical Research and Ethics Committee (reference number: ZAMREC/0002/JUN/17).

## 3. Results

Baseline demographic and clinical data were collected from a total of 488 participants (275 MNM cases and 213 controls) (see Figure 1). In brief, women who experienced MNM were more likely to be less educated (χ^2^ = 7.29, *p* < 0.007), had below-average wealth (χ^2^ = 10.11, *p* = 0.001) and higher parity (χ^2^ = 12.24, *p* < 0.001) compared with the controls (see Table 1 and Appendix A). Out of 488 participants, 342 (171 near-miss and 171 control) attended at least one follow-up visit (70% follow-up rate). There was no statistically significant difference between women who took part and women who were lost to follow-up in terms of baseline demographic and clinical characteristics (see Appendix A). The median (IQR) PHQ-9 score of all the three follow-up points was 1.0 for women that had MNM and the control group. The PHQ-9 score of nearly all women (99%) was less than the cutoff for moderate to severe depression, which is 10 points. Only 7% of the women had mild depression (scored between five and nine in PHQ-9). Participants scored relatively higher on the somatic symptoms of depression than mood and cognitive symptoms (see Appendix A). As shown in Table 2, the overall median (IQR) of HTQ-16 score of all the three follow-up points was 1.0 (1.0–1.1). The maximum HTQ-16 score was 2.4, indicating that all participants scored below the cutoff score for PTSD diagnosis of 2.5. Overall, participants scored high in all domains of QoL. The median score of all four domains of QoL ranges from 90.6 to 100. Participants scored relatively high in the psychological domain of QoL and relatively low in the environmental domain of QoL.

### 3.1. Mental Health Problems and QoL over Time

The depressive symptoms for women in the control group were stable over time and did not differ statistically from MNM women (Table 3). Even though PTSD symptoms were not statistically significantly different at the initial status (three-month) between MNM and control women (OR = 1.14, 95% CI = (0.72–1.78), the trajectory of PTSD symptoms was statistically significantly different between the two groups of women. For the control group, the odds ratio of having the minimum HTQ-16 score (HTQ-16 score of one) increased by a factor of 1.09 every month (OR = 1.09, 95% CI = (1.02–1.16)) while for near-miss women this time trend remained approximately stable (Appendix A).

Different domains of QoL showed different trajectories (see Table 4). First, the physical domain of QoL, MNM women scored 21% lower QoL at the initial status (three-month) compared with the controls, but their QoL increased monthly 5% faster than the controls. For controls, the QoL score remained stable over time and there were no statistically significant differences between MNM and control women at initial status (three-month) in the psychosocial, social as well as environmental domains of QoL.

### 3.2. Association of Risk Factors with Mental Health Problems and QoL

Full results from the mixed effects model analysis of risk factors with varying trajectories are presented in Table 3 and Table 4. Presence of intercurrent illness of the mother, perinatal loss, and having low social support was found to be statistically significantly associated with higher depressive symptoms. Controlling for other variables in the model, women with intercurrent illness scored 1.42 times higher on depressive symptoms (γ = 1.42, 95% CI = (1.15–1.75)) while women with perinatal loss had 1.30 times higher score of depressive symptoms than their counterparts (γ = 1.30, 95% CI = (1.01–1.67)). Likewise, when the social support scale score of a woman increases by one point, the PHQ-9 score decreased by 5% (γ = 0.95, 95% CI = (0.93–0.97)) (Table 3).

Apart from social support, which was associated with all domains of QoL, different risk factors were associated with different domains of QoL. A single unit increase in social support scale score resulted in a 7% increase in physical (γ = 0.93, 95% CI = (0.92–0.95)) and psychological (γ = 0.93, 95% CI = (0.90–0.97)) domain, 2% increase in social domain (γ = 0.98, 95% CI = (0.97–0.99)) and 9% increase in environmental domain of QoL (γ = 0.91, 95% CI = (0.90–0.92)). As illustrated in Table 4, the effect of social support on QoL was both on the zero-inflated as well as negative binomial part of the model. This means women with better social support had a higher probability to score 100 (maximum QOL) and their mean QoL score is higher compared with their counterparts.

Presence of intercurrent illness of the mother was associated with lower probability of a 100-score (OR = 0.41, 95% CI = (0.24–0.70)) and older age (γ = 1.01, 95% CI = (1.00–1.03)) was associated with lower average of physical domain of QoL score. Surprisingly, delivering the current child with cesarean section was associated with a higher probability of scoring 100 on physical domain of QoL compared with vaginal delivery (OR = 1.96, 95% CI = (1.21–3.19)), yet the mean QoL score of women with intercurrent illness and vaginal delivery was not statistically significantly different from their counterparts.

Having a history of a cesarean section was associated with a lower probability of a 100-score (OR = 0.35, 95% CI = (0.15–0.84)) in the psychological domain of QoL and unemployment (γ = 3.86, 95% CI = (1.34–11.15)) was associated with lower average of psychological domain of QoL.

Furthermore, perinatal loss was associated with a higher probability of scoring 100 in the social domain of QoL (OR = 1.62, 95% CI = (1.03–2.54)). Having an intercurrent illness of the mother (γ = 1.17, 95% CI = (1.05–1.30)) was associated with lower environmental domain QoL scores. Women with secondary or more education had a higher chance of getting a score of 100 in the environmental domain of QoL (OR = 2.36, 95% CI = (1.12–5.00)) (Table 4).

## 4. Discussion

This study was the first to examine changes over time and associated factors of mental health symptoms and QoL of women with severe acute obstetric complications (maternal near-miss) up to one-year post-partum in Zanzibar. Overall, we found that postpartum women reported low levels of mental health problems and high quality of life scores that did not change much over time, except for PTSD symptoms and physical QoL. Social support, perinatal loss and intercurrent illness were strong predictors of both mental health symptoms and QoL.

Interestingly, most post-partum women in this cohort in Zanzibar scored below the threshold for moderate depression, which is lower than the prevalence of depression in the general Tanzanian population [25]. Similarly, none of the participants had a PTSD diagnosis according to the cutoff point of the PTSD scale. The low reporting of women on depressive and PTSD symptoms might be due to several reasons. First, the type of scale or the concepts used in this study were validated in sub-Saharan Africa. However, the cultural expression and constructs of depression and PTSD might be different in Zanzibar, an island with own culture and traditions consisting predominantly Muslim population. Similar to other studies in Tanzania [25], we noticed that somatic symptoms of depression were more common than mood and cognitive symptoms among participants in our study. In addition, it might be that women were hesitant to report on symptoms knowing that mental illness is often associated with stigma and discrimination in SSA [26].

We found that severe maternal complications have a little effect on mental health and quality of life in our population irrespective whether they lost their baby or not. This finding is in line with previous studies done in Malaysia [10] and England [5] which equally did not find significant differences when screening for depression between women with and without severe maternal complications. However, there are other studies that found a statistically significant difference between the two groups [7,9], while another study [8] found that the association between depression and severe maternal morbidity was present only for women who lost their baby.

The finding that women with intercurrent illness and perinatal loss are at an increased risk of having more symptoms of depression is in line with several other studies. For example, Assarag et.al [7] found that perinatal death and serious illness were risk factors for depression among postnatal women. Similarly, Filippi et al. found that women with maternal complications and who lost their baby during the perinatal period were at increased risk of depression [8,9].

In this study, even though severe maternal complications started with a relatively lower physical QoL three months after the event of maternal complication, their QoL improved every month up to one year, which was not the case for women with-out complication. The lower initial physical QoL was likely to be caused by the effect of sever maternal complication (or the medical problem that caused the complication such as severe blood loss) on physical health. Nevertheless, the physical QoL of women with severe complication improved over time as these women physically recovered over time. Yet, the improvement in QoL over time might also be due to the phenomena known as “response shift” or “disability paradox”. Patients confronted with a life-threatening health condition are faced with the necessity to accommodate to their illness. This adaptation process may involve a change in one’s self-evaluation of position in life as a result of a change in internal standards (recalibration), value (reprioritization), or meaning (reconceptualization) [27]. For example, a woman’s demarcation of a good night sleep soon after the complication and a year after might be different.

Finally, women in this cohort reported high levels of social support, which was strongly related to lower depressive symptoms and associated with all domains of QoL. The positive effect of social support on QoL is almost universal [28]. This can be due to the “stress-buffering effect” of social support, meaning when stress levels are high, resources from people within the social environment facilitate coping, thus improving QoL. However, it can also be that the more social support an individual has, the better its QoL, regardless of stress level [29]. In particular, the amount of social support may be related to religious beliefs and norms by Zanzibar’s predominantly Muslim population, in which being socially supportive to each other is valued [30]. The two unpredicted findings of this study were women who lost their baby scored better social QoL and women who had cesarean delivery were more likely to have better physical QoL scores than their counterparts. This is a novel finding and future studies should look into it further. One possibility is these women may have received more support because of their situation of losing their baby and surgery than those women with a healthy baby and relatively normal delivery. Also, Islamic beliefs suggest that “hardship is a blessing”, which might partly contribute to participants’ resilience and coping with near-miss and loss of a baby [31].

### Limitations

There are some limitations to this study. First, the sample size was calculated to detect at least a medium effect size. Therefore, it is possible that our study missed detecting associations between variables that were present with a smaller effect size. Future studies with a larger sample size are therefore required. Second, there were dropouts in this study, however, analysis of the characteristics of the participants and dropouts showed that there was no difference suggesting that the missing values were at random. Third, the selected questionnaires were not validated previously in the Zanzibarian context. Adapting instruments to the local language and culture should make sure not only the linguistic equivalence, but also the conceptual equivalence, ensuring same underlying construct is being measured, and technical equivalence, ensuring that the methods of administration are comparable to avoided instrument bias [32]. Future studies should mix qualitative and quantitative design to better elicit the cultural expression of mental health problems and QoL. Fourthly, outcome variables were measured only during three, six, and twelve-month follow-up, but not at baseline. As a result, results could only be compared with controls but not with their initial values. Lastly, other factors that were not measured in our study such as history of previous pregnancy loss and nutritional status may have influenced the outcomes.

## 5. Conclusions

In this cohort study, we found that postpartum women reported low scores on depressive, PTSD symptoms, and high scores on QoL from the period from delivery to the first year postpartum. Social support was strongly associated with better mental health as well as quality of life. Therefore, we propose integration of women’s social networks in preventing and managing the aftermath of maternal complications, in particular in the case of intercurrent illness and perinatal death. Further studies with larger sample size, mixing qualitative and quantitative design, and scales adapted to local language and culture are required.

## Figures and Tables

**Figure 1 ijerph-17-09034-f001:**
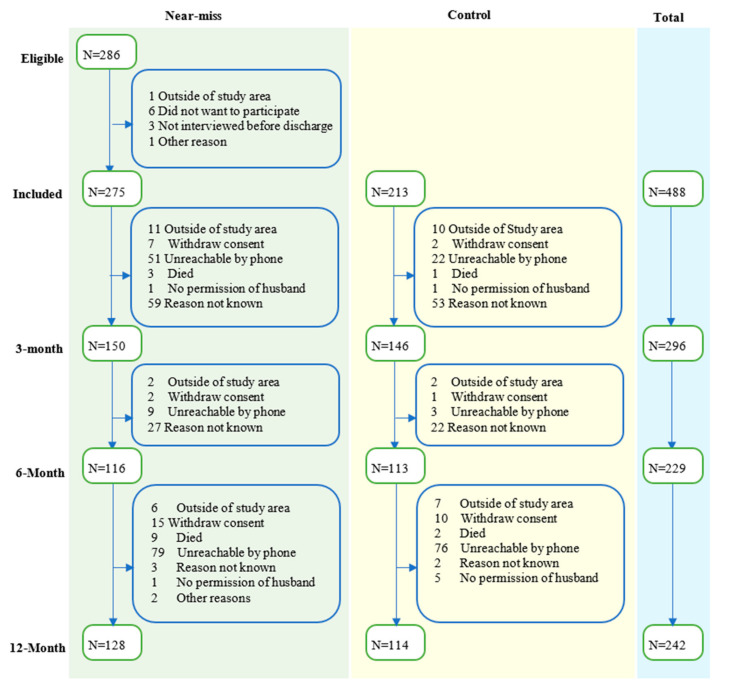
Flowchart of eligibility, inclusion, and follow-up of participants.

**Table 1 ijerph-17-09034-t001:** Baseline demographic and clinical characteristics of participants included in the study ^¥.^

Variable	Category	MNM (N = 275)	Control (N = 213)	Total (N = 488)	*p*-Value ^§^
Age	Mean (SD)	29.1 (6.0)	28.3 (5.6)	28.8 (5.9)	0.080
Marital Status	Married	201 (91.0)	184 (92.9)	385 (91.9)	0.459
	Unmarried	20 (9.0)	14 (7.1)	34 (8.1)	
Residence	Rural	32 (29.9)	23 (34.3)	55 (31.6)	0.476
	Urban	33 (30.8)	15 (22.4)	48 (27.6)	
	Mixed	42 (39.3)	29 (43.3)	71 (40.8)	
Education	Primary or less	90 (42.1)	57 (29.2)	147 (35.9)	0.007 *
	Secondary or more	124 (57.9)	138 (70.8)	262 (64.1)	
Employment	Employed	30 (13.9)	34 (17.3)	64 (15.5)	0.269
	Housewife	121 (56.0)	98 (49.7)	219 (53.0)	
	Self-employed	58 (26.9)	52 (26.4)	110 (26.6)	
	Others ^†^	7 (3.2)	13 (6.6)	20 (4.8)	
Perceived Wealth	Average	160 (76.2)	168 (88.4)	328 (82.0)	0.001 *
	Below average	50 (23.8)	22 (11.6)	72 (18.0)	
Gestational age	First trimester	17 (9.4)	19 (13.5)	36 (11.2)	0.272
	Second trimester	12 (6.7)	5 (3.5)	17 (5.3)	
	Third trimester	151 (83.9)	117 (83.0)	268 (83.5)	
Mode of termination of pregnancy	Vaginal delivery	74 (28.1)	60 (28.8)	134 (28.4)	0.827
	Cesarean section	146 (56.1)	116 (55.8)	262 (55.4)	
	Early pregnancy loss	37 (14.1)	25 (12.0)	62 (13.2)	
	Still pregnant at discharge	7 (2.7)	7 (3.4)	14 (3.0)	
Parity	1	100 (38.6)	113 (53.3)	213 (45.2)	0.002 *
	2–4	105 (40.5)	74 (34.9)	179 (38.0)	
	>4	54 (20.8)	25 (11.8)	79 (16.8)	
History of cesarean section	Yes	40 (27.8)	18 (19.6)	58 (24.6)	0.153
	No	104 (72.2)	74 (80.4)	178 (75.4)	
Pregnancy outcome	Livebirth	73 (42.9)	125 (74.0)	198 (58.4)	<0.001 *
	Early pregnancy loss	26 (15.3)	21 (12.4)	47 (13.9)	
	Perinatal loss	71 (41.8)	23 (13.6)	94 (27.7)	

^¥^ Quantities in the table indicate Number (%), except for age, which indicates Mean (SD). ^§^
*p*-value for a univariate association of demographic and clinical characteristics with near-miss status. ^†^ Students, unemployed and non-paid job. * *p*-value less than 0.05.

**Table 2 ijerph-17-09034-t002:** Descriptive results of outcome variables over time for maternal near-miss versus control women.

	3 Months		6 Months		12 Months		Total
	MNM	Controls	MNM	Controls	MNM	Controls	
Mental Health-Median (IQR)							
PHQ-9 Score	1.0 (0.0–3.0)	1.0 (0.0–2.0)	1.0 (0.0–2.0)	1.0 (0.0–2.0)	1.0 (0.0–3.0)	1.0 (0.0–2.0)	1.0 (0.0–2.0)
HTQ Score	1.0 (1.0–1.1)	1.0 (1.0–1.1)	1.0 (1.0–1.1)	1.0 (1.0–1.1)	1.0 (1.0–1.1)	1.0 (1.0–1.1)	1.0 (1.0–1.1)
WHOQOL-BREEF Domain Scores-Median (IQR)							
Physical	85.7 (71.4–96.4)	89.3 (78.6–100.0)	92.9 (78.6–96.4)	92.9 (78.6–100.0)	92.9 (89.3–100.0)	96.4 (89.3–100.0)	92.9 (82.1–100.0)
Psychological	95.8 (79.2–100.0)	95.8 (79.2–100.0)	95.8 (91.7–100.0)	100.0 (83.3–100.0)	100.0 (95.8–100.0)	100.0 (95.8–100.0)	100.0 (91.7–100.0)
Social	75.0 (58.3–100.0)	83.3 (58.3–100.0)	100.0 (75.0–100.0)	100.0 (58.3–100.0)	100.0 (91.7–100.0)	100.0 (93.8–100.0)	100.0 (66.7–100.0)
Environmental	87.5 (75.8–93.8)	87.5 (75.0–93.8)	90.6 (84.4–93.8)	90.6 (78.1–93.8)	93.8 (87.5–93.8)	93.8 (90.6–96.9)	90.6 (84.4–93.8)

**Table 3 ijerph-17-09034-t003:** Results of the negative binomial linear mixed-effects model for depressive symptoms ^a,b^.

Variable	EXP (Estimate)	95% CI		*p*-Value
		Lower	Upper	
Intercept	5.03	2.67	9.49	0.000
Time after discharge from hospital	1.00	0.96	1.03	0.888
MNM status × time after discharge	1.01	0.96	1.06	0.705
MNM status (near-miss women)	1.05	0.79	1.38	0.750
Pregnancy outcome (Early pregnancy loss)	0.91	0.66	1.26	0.582
Pregnancy outcome (Perinatal loss)	1.30	1.01	1.67	0.041
Intercurrent illness of mother (yes)	1.42	1.15	1.75	0.001
Social support scale	0.95	0.93	0.97	0.000

^a^ The reference category for variables in the model is as follows: maternal near miss (MNM) status (control women), pregnancy outcome (baby alive), and intercurrent illness of mother (No). ^b^ Estimates and confidence intervals are exponentiated. The un-exponentiated (log scale) outputs can be found on Appendix A.

**Table 4 ijerph-17-09034-t004:** Results of the negative binomial mixed-effects model for four domains of quality of life ^a,b,c,d.^.

	Negative Binomial				Zero-Inflated			
	EXP (Estimate)	95% CI		*p*-Value	EXP (Estimate)	95% CI		*p*-Value
		Lower	Upper			Lower	Upper	
	Physical Domain of Quality of Life							
(Intercept)	77..89	45.18	134.30	0.000	0.00	0.00	0.01	0.000
Time after discharge from hospital	1.00	0.98	1.03	0.888	1.01	0.95	1.09	0.696
MNM status (near-miss women)	1.21	1.01	1.45	0.038	0.62	0.34	1.11	0.107
MNM status ×time after discharge	0.95	0.92	0.99	0.007	1.00	0.90	1.10	0.955
Intercurrent illness of mother (yes)	1.11	0.96	1.29	0.152	0.41	0.24	0.70	0.001
Social support scale	0.93	0.92	0.95	0.000	1.31	1.16	1.48	0.000
Mode of delivery (Early pregnancy loss)	1.01	0.82	1.25	0.892	1.39	0.73	2.67	0.317
Mode of delivery (Cesarean section)	0.89	0.76	1.03	0.120	1.96	1.21	3.19	0.006
Age of the women	1.01	1.00	1.03	0.029	0.96	0.93	1.00	0.063
	Psychological Domain of Quality of Life							
(Intercept)	93.06	26.52	326.55	0.000	0.00	0.00	0.07	0.003
Time after discharge from hospital	1.00	0.92	1.08	0.978	0.93	0.80	1.08	0.339
MNM status (near-miss women)	1.42	0.80	2.51	0.228	0.58	0.21	1.62	0.301
MNM status × time after discharge	0.93	0.85	1.03	0.179	1.11	0.93	1.33	0.243
Intercurrent illness of mother (yes)	1.00	0.65	1.52	0.987	0.62	0.30	1.28	0.196
History of C-Section (Only one)	1.09	0.70	1.69	0.711	0.35	0.15	0.84	0.018
History of C-Section (two or more)	0.83	0.38	1.82	0.647	0.87	0.22	3.36	0.834
Social support scale	0.93	0.90	0.97	0.001	1.34	1.13	1.59	0.001
Residence (Mixed)	0.64	0.40	1.01	0.053	1.33	0.59	2.97	0.493
Residence (Rural)	0.84	0.55	1.29	0.423	0.69	0.30	1.59	0.390
Employment (Housewife)	1.05	0.64	1.73	0.841	1.67	0.67	4.13	0.269
Employment (Self-employed)	1.20	0.70	2.09	0.507	1.16	0.41	3.29	0.778
Employment (Unemployed) ^d^	3.86	1.34	11.15	0.013	2.20	0.40	12.15	0.365
	Social Domain of Quality of Life							
(Intercept)	66.71	49.82	89.31	0.000	0.00	0.00	0.01	0.000
Time after discharge from hospital	0.99	0.97	1.02	0.464	1.12	1.04	1.20	0.002
MNM status (near-miss women)	1.02	0.88	1.18	0.833	0.75	0.45	1.26	0.280
MNM status × time after discharge	0.98	0.95	1.01	0.180	0.98	0.89	1.08	0.689
Pregnancy outcome (Early pregnancy loss)	0.88	0.74	1.03	0.113	1.24	0.72	2.15	0.438
Pregnancy outcome (Perinatal loss)	1.00	0.87	1.15	0.991	1.62	1.03	2.54	0.035
Education (Secondary or more)	0.92	0.82	1.03	0.152	1.39	0.94	2.06	0.096
Social support scale	0.98	0.97	0.99	0.000	1.21	1.14	1.29	0.000
	Environmental Domain of Quality of Life							
(Intercept)	229.64	166.83	316.10	0.000	0.00	0.00	0.02	0.001
Time after discharge from hospital	0.97	0.95	0.99	0.001	1.03	0.94	1.14	0.482
MNM status (near-miss women)	1.02	0.90	1.17	0.737	0.92	0.41	2.11	0.851
MNM status × time after discharge	1.01	0.99	1.03	0.432	0.87	0.74	1.02	0.085
Intercurrent illness of mother (yes)	1.17	1.05	1.30	0.004	0.60	0.27	1.32	0.205
Social support scale	0.91	0.90	0.92	0.000	1.24	1.04	1.47	0.014
Education (Secondary or more)	0.94	0.84	1.04	0.216	2.36	1.12	5.00	0.024

^a^ Scores of all domains of QoL are reversely coded in these models, which means zero indicates maximum and 100 the minimum QoL. ^b^ The reference category for variables in the model are as follows: MNM status (control women), intercurrent illness of mother (no), mode of delivery (vaginal delivery), history of cesarean section (no previous cesarean section), address (rural), employment (employed), perinatal loss (no), perceived wealth (average), and education (primary or less). ^c^ Estimates and confidence intervals are exponentiated. The un-exponentiated (log scale) outputs can be found on Appendix A. ^d.^ Also include students and non-paid job.

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
