# Peer review of "Mental Health and Quality of Life of Women One Year after Maternal Near-Miss in Low and Middle-Income Countries: The Case of Zanzibar, Tanzania"

_ijerph, 2020, doi:10.3390/ijerph17239034_

Round 1

Reviewer 1 Report

Dear Authors,

May be the title of your article should add “low and middle-income countries women”

Line 60 you write: “A research assistant invited all eligible women in the hospital during the study period who fulfilled inclusion criteria to participate in the study.” Did you obtained an informed consent to participate in the study, ...? Please specify.

Line 76-78, you write: “All questionnaires, except PHQ-9, were translated by three research assistants who are health-care workers, and fluent both in Kiswahili and English, into Kiswahili and back-translated to English” we suggest a better explanation of the utility of those two direction translations with some references, and on how the questionnaires were administrated (with or without the presence and help of the assistants).

Other factors may have influenced the results of you study like the nutritional status of the participants, the presence or not of infection diseases along the study, historical of previous pregnancy loss experience, maybe it should be of interest to discuss some of those point or consider as limits of the study?

Thank you for your attention and best wishes for the publication of your paper.

Author Response

Reviewer 1

May be the title of your article should add “low and middle-income countries women”

Thank you for your suggestion. We now have changed the title as follows: Mental health and quality of life of women one year after maternal near-miss in low and middle-income countries: the case of Zanzibar, Tanzania

Line 60 you write: “A research assistant invited all eligible women in the hospital during the study period who fulfilled inclusion criteria to participate in the study.” Did you obtained an informed consent to participate in the study, ...? Please specify.

Thank you for your valid comment. We now added the following sentence in Page 2, line 70. … … …“Inclusion in the study took place after written (or oral in case of illiteracy) informed consent was obtained from each participant.

Line 76-78, you write: “All questionnaires, except PHQ-9, were translated by three research assistants who are health-care workers, and fluent both in Kiswahili and English, into Kiswahili and back-translated to English” we suggest a better explanation of the utility of those two direction translations with some references, and on how the questionnaires were administrated (with or without the presence and help of the assistants).

This is a relevant comment to increase the articulation and readability of our paper. As you recommended, we included a better explanation and references. We hope we made it more clear now. (page 3 line 94) …”Translation was performed using the World Bank three step questionnaire translation: 1) forward translation (one research assistant translated the original scale to Kiswahili), 2) back translation (a different research assistant translates the Kiswahili version back to English) and 3) reconciliation (the three research assistants together compared the original scale to the back translated questionnaire, discuss all discrepancies and agree on a final translation) [21].”

Other factors may have influenced the results of you study like the nutritional status of the participants, the presence or not of infection diseases along the study, historical of previous pregnancy loss experience, maybe it should be of interest to discuss some of those point or consider as limits of the study?

Thank you for your feedback. We agree and have added the following limitation (page 11, line 285). …” Lastly, other factors that were not measured in our study such as history of previous pregnancy loss and nutritional status may have influenced the outcomes.”.

Thank you for your attention and best wishes for the publication of your paper.

Thank you so much for all your constructive comments!

Reviewer 2 Report

Thank you for the opportunity to review this paper reporting the mental health and quality of life of Islamic women in Zanzibar for a year following a maternal near miss. A prospective case-control study was undertaken using previously validated tools to assess symptoms of depression and post-traumatic stress disorder, quality of life and perceived social support. The key findings were that “postpartum women reported low levels of mental health problem and high quality of life scores which did not change much over time.” Social support, perinatal loss and inter-current illness were associated with both depressive symptoms and quality of life.

The methods used are appropriate, and the interpretation of the results appears to be valid.

The introduction is sparse, one paragraph and the related literature on the subject is summarised as “results are inconclusive” with nine papers cited in support of this. The paper would be much improved by a summary of the relevant literature from both the developed and developing world. Similarly the discussion is short, three paragraphs, and it too should be expanded. The findings that maternal near miss has little effect on mental health and quality of life in this population should be contextualised within the existing literature.

The Quality of Life data and depressive symptoms were right and left skewed, respectively. To overcome this, the modelling used negative binomial distribution. As I am not conversant with this method, I cannot comment on its validity.

Ethical approval for the study was gained, but no mention is made of participants consenting to be part of the study, other than 37 people withdrawing their consent during the study. Is there any explanation for this?

Figure 1: to what does the word “unknown” refer in figure one. Roughly a third of the participants lost to study are classified as “unknown”.  

Line 206: there is a superscript 36 after South Africa.

Author Response

Reviewer 2

Thank you for the opportunity to review this paper reporting the mental health and quality of life of Islamic women in Zanzibar for a year following a maternal near miss. A prospective case-control study was undertaken using previously validated tools to assess symptoms of depression and post-traumatic stress disorder, quality of life and perceived social support. The key findings were that “postpartum women reported low levels of mental health problem and high quality of life scores which did not change much over time.” Social support, perinatal loss and inter-current illness were associated with both depressive symptoms and quality of life.

The methods used are appropriate, and the interpretation of the results appears to be valid. 

Thank you very much for your commendation. This is very motivating.

The introduction is sparse, one paragraph and the related literature on the subject is summarised as “results are inconclusive” with nine papers cited in support of this. The paper would be much improved by a summary of the relevant literature from both the developed and developing world.

Thank you for the comment. We now added more details about the studies including a summary of the relevant literature (page 2, line 49). ….”For instance, most studies (e.g., conducted in Morocco [7], Benin [8], and Burkina Faso [9]) found that women with MNM had an increased risk of developing depression in the post-partum year, especially if they had lost their baby. On the contrary, similar studies in Malaysia [10] and England [5] found no significant difference regarding the risk of depression among women with and without MNM. Similarly, there is contradicting evidence regarding the association between hospital admission due to pregnancy complications and Post Traumatic Stress Disorder (PTSD) [5, 6, 11]. This study aims to investigate mental health (depression and PTSD) and QoL of these women over time.”

Similarly the discussion is short, three paragraphs, and it too should be expanded.

We totally agree with you. We have expanded the discussion as per your feedback. Please see page 10-11. Thank you for your comment!

The findings that maternal near miss has little effect on mental health and quality of life in this population should be contextualised within the existing literature.

Thank you for your feedback. We have expanded this section by discussing the finding in the context of existing literature (page 10, line 231). … “We found that severe maternal complications have a little effect on mental health and quality of life in our population. This finding is in line with previous studies done in Malaysia [10] and England [5] which equally did not find significant differences when screening for depression between women with and without severe maternal complications. However, there are other studies that found a statistically significant difference between the two groups [7, 9], while another study [8] found that the association between depression and severe maternal morbidity was present only for women who lost their baby.”

Ethical approval for the study was gained, but no mention is made of participants consenting to be part of the study, other than 37 people withdrawing their consent during the study. Is there any explanation for this?

Thank you for your valid comment. We now, made it more clear by including the following sentence in Page 2, line 70. …… “Inclusion in the study took place after written (or oral in case of illiteracy) informed consent was obtained from each participant.”

Figure 1: to what does the word “unknown” refer in figure one. Roughly a third of the participants lost to study are classified as “unknown”.  

The word “unknown” meant the reason for dropout not known. We now replaced it with “Reason not known” on the entire figure. Hope it is more clear now.

Line 206: there is a superscript 36 after South Africa.

Apologies for the typing error. It is corrected now.

Reviewer 3 Report

This is a paper on  mental health and quality of life trajectories of women who had a near-miss during or after pregnancy.

  1. The "results" section is difficult to read and needs to be semplified
  2. Women with psychosis have been excluded because psychosis may affect history taking. I suggest authors to read this paper: A comparison between self-report and interviewer-rated retrospective reports of childhood abuse among individuals with first-episode psychosis and population-based controls. Gayer-Anderson C, Reininghaus U, Paetzold I, Hubbard K, Beards S, Mondelli V, Di Forti M, Murray RM, Pariante CM, Dazzan P, Craig TJ, Fisher HL, Morgan C.J Psychiatr Res. 2020 Apr;123:145-150. doi: 10.1016/j.jpsychires.2020.02.002. Epub 2020 Feb 6.PMID: 32065950
  3. In the paper it is stated that "QoL increased monthly 5% faster than controls" (line 156): this could be due to regression towards the mean and should be discussed

Author Response

Reviewer 3

This is a paper on  mental health and quality of life trajectories of women who had a near-miss during or after pregnancy.

  1. The "results" section is difficult to read and needs to be simplified

Thank you for your feedback. Indeed, it is complicated to explain the results because the outcome variables were very skewed and had to be fitted with complex models, which are also difficult to explain and interpret. We acknowledge that the results are difficult to understand when not being familiar with the zero-inflated and negative binomial distributions, but we did our best to highlight the most important results as simple as possible. Unfortunately, more simple models are not suitable (and even not possible to estimate).

  1. Women with psychosis have been excluded because psychosis may affect history taking. I suggest authors to read this paper: A comparison between self-report and interviewer-rated retrospective reports of childhood abuse among individuals with first-episode psychosisand population-based controls. Gayer-Anderson C, Reininghaus U, Paetzold I, Hubbard K, Beards S, Mondelli V, Di Forti M, Murray RM, Pariante CM, Dazzan P, Craig TJ, Fisher HL, Morgan C.J Psychiatr Res. 2020 Apr;123:145-150. doi: 10.1016/j.jpsychires.2020.02.002. Epub 2020 Feb 6.PMID: 3206595

Thank you for your feedback and for sharing this nice study. However we found it too risky to put that to practice in our study context, in which there hasn’t been a comparable study like the one of Gayer-Anderson et al., neither is there a comprehensive, easily accessible multi-level mental health care system in place, to swiftly address when needed.

  1. In the paper it is stated that "QoL increased monthly 5% faster than controls" (line 156): this could be due to regression towards the mean and should be discussed

This is interesting. We think there is a logical explanation and would not interpret this effect as regression to the mean. We believe it makes sense to think the near-miss experience makes women to have lower physical QoL at the beginning and later they recover to their usual state of quality of life (at the last waves they are more similar to the control values). We now included this in discussion (please see page 11, line 243 and page 11, 283)

Reviewer 4 Report

Overall this is an interesting manuscript which although well-written, still requires proofreading and spell checks. the presentation of the manuscript is not uniform and seems like there are two or more contributors to the manuscript, drawing from the different styles of presentation of the results. for example, one of the authors uses full-stop while the other uses commas (for example, compare Table 1 and Table 4). I recommend streamlining the presentation of results. Additionally in the tables, there is no key and it is not clear what the figures in the tables represent for example, see Table 1-what do the figures in the brackets and outside the brackets represent. There is a spelling mistake in the last line on page 141, that is, 'non-payed job', do the authors mean 'non-paying job' or non-paid job?

Consider defining the different variables and what the different measures mean in the Methods section. include the cut-off points in the methods section and not in the results section (page 135).

Author Response

Reviewer 4

Overall this is an interesting manuscript which although well-written, still requires proofreading and spell checks. The presentation of the manuscript is not uniform and seems like there are two or more contributors to the manuscript, drawing from the different styles of presentation of the results. for example, one of the authors uses full-stop while the other uses commas (for example, compare Table 1 and Table 4). I recommend streamlining the presentation of results.

Thank you very much for your feedback! We believe it is important comment to increase the readability of our paper. As you recommended, we thoroughly copyedited the whole manuscript.

Additionally in the tables, there is no key and it is not clear what the figures in the tables represent for example, see Table 1-what do the figures in the brackets and outside the brackets represent.

Thank you for your comment! We indicated in the footnote of the table that “Quantities in the table indicate Number (%), except for age, which indicates Mean (SD)”

There is a spelling mistake in the last line on page 141, that is, 'non-payed job', do the authors mean 'non-paying job' or non-paid job?

Apologies for the typing error. It is corrected now.

Consider defining the different variables and what the different measures mean in the Methods section. Include the cut-off points in the methods section and not in the results section (page 135).

Thank you for your comment! Based on your recommendation, we defined each outcome variable now and included a cutoff point in the method section (see page 2, line 82). ….. “PHQ-9 is a depression symptom scale, which rates each of the nine Diagnostic and Statistical Manual (DSM-IV) criteria on a four npoint Likert scale. PHQ-9 scores of 5, 10, 15, and 20 represents mild, moderate, moderately severe, and severe depression, respectively [15]. The PHQ-9 has been translated and validated in Tanzania [16]. HTQ-16 screens for PTSD using 16 trauma symptom items derived from the DSM-IV PTSD criteria, rating each item on a four point Likert scale. The HTQ total score is an average score, with 2.5 and above indicating a high likelihood of PTSD [17].

Round 2

Reviewer 3 Report

Thank you for your revision. 

I still think you should explain why you did not include in the study people with psychosis. They have been proved to be reliable in self-reporting traumatic events. The attitude of excluding them can contribute to stigma and this should be avoided.